# Evaluation of the fatty acid-based erythrocyte membrane lipidome in cats with food responsive enteropathy, inflammatory bowel disease and low-grade intestinal T-cell lymphoma

**Paolo Emidio Crisi** [1]*, **Maria Veronica Giordano**[1], **Alessia Luciani**[1],
**Alessandro Gramenzi**[1], **Paraskevi Prasinou**[1], **Anna Sansone**[2], **Valentina Rinaldi**[1],
**Carla Ferreri**[2], **Andrea Boari**[1]

**1** Department of Veterinary Medicine, Veterinary Teaching Hospital, University of Teramo, Teramo, Italy,
**2** Institute of Organic Synthesis and Photoreactivity, Consiglio Nazionale delle Ricerche, Bologna, Italy

* pecrisi@unite.it

## Abstract

Feline chronic enteropathies (FCE), include food-responsive-enteropathy (FRE), inflammatory bowel disease (IBD), and low-grade intestinal T-cell lymphoma (LGITL), and are common causes of chronic gastrointestinal signs in cats. Distinguishing between different subgroups of FCE can be challenging due to the frequent overlap of anamnestic, clinical, and laboratory data. While dysregulation in lipid metabolism has been reported in humans and dogs with chronic IBD, similar changes in cats are not yet completely understood. Assessing the fatty acid (FA) profile of red blood cell (RBC) membranes offers a valuable method for evaluating the quantity and quality of structural and functional molecular components in the membranes. Therefore, this study aimed to examine the FA composition of RBC membranes in FCE in comparison to healthy cats (HC). Gas-chromatography was used to quantitatively analyze a cluster of 11 FA, and based on these results, parameters of lipid homeostasis and enzyme activity indexes were calculated. A total of 41 FCE cats (17 FRE, 15 IBD, 9 LGITL) and 43 HC were enrolled. In FCE cats, the values of docosapentaenoic acid (p = 0.0002) and docosahexaenoic acid (p = 0.0246), were significantly higher, resulting in an overall increase in ω-3 polyunsaturated fatty acids (PUFA) (p = 0.006), and that of linoleic acid (p = 0.0026) was significantly lower. Additionally, FCE cats exhibited an increased PUFA balance (p = 0.0019) and Δ6-desaturase index (p = 0.0151), along with a decreased ω-6/ω-3 ratio (p = 0.0019). No differences were observed among cats affected by FRE, IBD and LGITL. Like humans and dogs, the results of this study indicate that FCE cats also display changes in their FA lipid profile at the level of the RBC membrane. The non-invasive analysis of RBC membrane shows promise as a potential tool for gaining a better understanding of lipid imbalances in this disease.

**Data Availability Statement:** All relevant data are within the paper and its Supporting information files.

**Funding:** The author(s) received no specific funding for this work.

**Competing interests:** The authors have declared that no competing interests exist.

## Introduction

Feline chronic enteropathy (FCE) is a common gastrointestinal condition in cats, defined by persistent or intermittent GI signs of at least 3 weeks of duration, associated with intestinal inflammation, in the absence of extra-intestinal causes or infectious, obstructive, or localized neoplastic intestinal diseases [1]. These disorders are commonly classified based on treatment response into food-responsive-enteropathy (FRE), inflammatory bowel diseases (IBD) or steroid-responsive enteropathies (SRE), and low-grade intestinal T-cell lymphoma (LGITL) [1–6].

While the recognition of FRE may be easier, as it presents usually a complete clinical resolution in response to dietary intervention, distinguishing between IBD and LGITL can be challenging due to the overlapping of anamnestic, clinical, and laboratory data between the two pathologies. Current diagnosis and differentiation require complex and more invasive procedures like histopathological examination of intestinal tissue biopsies and often additional diagnostic test such as immunohistochemistry and clonality testing. Treatment often involves the use of immunosuppressive drugs [4–6]. Research efforts are focused on exploring new and less invasive diagnostic and prognostic biomarkers, as well as identifying potential therapeutic targets to improve FCE diagnosis and management.

Digestion and turnover of lipids are essential functions of the gastrointestinal tract. Lipids contain fatty acids that play different roles as major building blocks for cell membrane phospholipids, fundamental energy sources, hormones, and signaling molecules [7]. Many studies report that lipid metabolism is impaired in humans with IBD as well as in animal models of IBD [8–12]. Some clinical data suggest that phospholipids, in addition to their role in membrane formation, have many further metabolic roles (regulation of gene expression, regulation of lipid and glucose homeostasis, steroid biosynthesis, cell proliferation, and alteration of membrane's receptors function by altering membrane fluidity [7]. Fatty acids (FAs) need to maintain their diversity, namely the appropriate presence of saturated and unsaturated structures to keep the structural properties and functional properties of cell membrane phospholipids, as well as play different roles in several other cellular compartments [7]. Studies in human medicine have shown that FAs play an important role in various phases of the inflammatory process. These include the pathogen recognition phase, in which pathogens penetrate the epithelial barrier or bond to receptors in regulating gene expression; the mobilization phase, in which immune cells immigrate from blood to the affected tissue, and the resolution phase, in which harmful agents are eliminated by anti-inflammation mediators [13]. The balance between ω-6 and ω-3 polyunsaturated FAs (PUFAs) and the ratio of saturated and unsaturated FAs are therefore crucial in regulating signaling outcomes and membrane properties [14–16].

The FA profiles have been investigated in different biological samples, such as serum, whole blood, urine, tissues, and stools, using different techniques in human patients with ulcerative colitis (UC) and Chron Disease (CD) [8, 16–22].

Recently, a FA-based RBC membrane lipidomic analysis was conducted in dogs with chronic enteropathy (CE), which revealed significant differences in the lipidomic profile of erythrocytes between dogs with CE and healthy dogs, such as higher levels of stearic acid, dihomo-gamma-linolenic (DGLA), eicosapentaenoic acid (EPA), and docosahexaenoic acid (DHA), and lower levels of palmitic and linoleic acids in CE dogs [23].

To date, knowledge is limited about the lipidomic profile of RBC membranes in cats and the authors hypothesize that the RBC membrane lipidome may mirror the "gut health" of cats affected by chronic enteropathy, as already observed in dogs. A recent study demonstrated the presence of lipid maldigestion and malabsorption in FCE [24] and the understanding of the lipid alterations can provide insights into the FCE mechanisms with potential future

therapeutic applications. For these reasons the present study aimed to investigate the differences in the quantity and quality of RBC membrane FAs between healthy cats and those with chronic gastrointestinal diseases.

## Material and methods

### Study population and diagnostic investigations

Cats admitted to the Veterinary Teaching Hospital (VTH) of the University of Teramo between January 2018 and November 2022 were prospectively enrolled in the study. The project was approved by the Committee on Animal Research and Ethics of the Universities of Chieti-Pescara, Teramo, and Experimental Zooprophylactic Institute of AeM (CEISA), Protocol UNICHD12 n. 1168. Prior to enrollment in the study, cat owners provided written informed consent.

Cats with persistent or intermittent clinical signs of chronic enteropathy lasting for at least 3 weeks, were eligible for enrollment in the group of cats with chronic enteropathy (FCE). Extra-gastrointestinal disease, infectious and parasitic intestinal diseases and localized neoplastic intestinal lesions were excluded based on a history, physical examination (including a 9-point BCS), complete blood count (CBC), serum biochemistry profile (i.e. glucose, blood urea nitrogen, creatinine, total bilirubin, aspartate aminotransferase, alanine aminotransferase, γ-glutamyl transferase, serum alkaline phosphatase, creatin kinase, DGGR lipase, calcium, phosphorus, albumin, total proteins, cholesterol, triglycerides, sodium, potassium, chloride, magnesium), urinalysis, total T4 (for cats > 6 years of age), feline serum pancreatic lipase (fPLI) and feline trypsin-like immunoreactivity (fTLI), a direct fecal smear evaluation and zinc sulfate centrifugal flotation, and abdominal ultrasound.

Exclusion criteria were antiacids or antibiotic treatment in the previous 6 months, and the presence of systemic or extra-gastrointestinal disease. Also, cats that have received dietary ω-3 supplementation in the last 4 months were not included in the present study.

In FCE cats, when available, serum cobalamin and folate concentrations, FCE activity index (FCEAI) [25] were recorded.

In the same period, examined cats without clinical or clinic-pathological evidence of disease, determined by a comprehensive assessment that included medical history, physical examination, CBC, serum biochemistry and urinalysis, were included as healthy cats (HC).

FCE cats were classified as having food-responsive enteropathy (FRE), idiopathic inflammatory bowel disease (IBD), and low-grade intestinal T-cell lymphoma (LGITL) [1–3]. In particular, the group of FRE includes patients who experienced complete remission of gastrointestinal symptoms within 3 weeks of a dietetic trial with a single novel protein source or hydrolyzed protein diet [26]. The remission was considered complete if clinical signs were resolved or the FCEAI score was reduced by ≥75% after three weeks of dietary therapy [25]. Patients who failed to respond to at least two dietary trials underwent gastro-duodenoscopy and ileo-colonoscopy for diagnostic purposes and biopsy specimens were evaluated by board-certified pathologists.

When an underlying LGITL was suspected by the pathologist based on histopathology, additional CD3 and CD20 immunohistochemical staining and clonality test [4–6] were performed to confirm the diagnosis. A final diagnosis of IBD or LGITL was reached by integrating the results from histopathology, immunohistochemistry, and clonality test. The same therapeutic diet was continued throughout the trial in all FCE cats.

Moreover, according to the folates and cobalamin levels, FCE cats were classified as hypofolatemic, normofolatemic or hyperfolatemic (reference interval 9.7–21.6 μg/L), and

**Table 1. Cluster of individual fatty acids determined in study cats and related families.** Together with the common names, the abbreviations describe the position and geometry of the double bonds (e.g., 9c for palmitoleic acid), as well as the notation of the carbon chain length and total number of double bonds (e.g., C18:1).

| Families of Fatty Acids | | Individual Fatty Acids |
|---|---|---|
| Saturated Fatty Acids | | Palmitic acid (C16:0) |
| | | Stearic acid (C18:0) |
| Monounsaturated Fatty Acids | | Palmitoleic Acid (C16:1) |
| | | Oleic Acid (9c, C18:1) |
| | | Vaccenic Acid (11c, C18:1) |
| Polyunsaturated Fatty Acids | ω-6 Polyunsaturated Fatty Acids | Linoleic Acid (C18:2) |
| | | Dihomo-gamma-linolenic Acid (C20:3) |
| | | Arachidonic Acid (AA, C20:4) |
| | ω-3 Polyunsaturated Fatty Acids | EPA (C20:5) |
| | | DPA (C22:5) |
| | | DHA (C22:6) |

EPA: eicosapentaenoic acid; DPA: docosapentaenoic acid; DHA: docosahexaenoic acid (DHA),

hypocobalaminemic, normocobalaminemic and hypercobalminemic (reference interval 290–1500 ng/L).

## Fatty acid-based erythrocyte membrane lipidome analysis

At the time of the admission, from each cat, blood samples were collected from the jugular vein using EDTA tubes.

The isolation of phospholipids from RBC membranes and transesterification procedure, gas-chromatographic analysis of fatty acid methyl esters (FAME) and calibration procedure were performed as previously described for dogs [23, 27].

Gas-chromatography has a flame- ionization detector (FID) which allows to quantify the fatty acids by having standard references for each of the fatty acid of the mixture.

A cluster of 11 FAs, representative of the main FA moieties present in the cell membrane, was chosen (Table 1). In particular, the cohort of FAs included: palmitic (C16:0) and stearic (C18:0) acids as SFAs; palmitoleic (C16:1), oleic (9c, C18:1), and vaccenic (11c, C18:1) acids as MUFAs; linoleic (LA, C18:2), DGLA (C20:3), and arachidonic (AA, C20:4) acids as ω-6 PUFA; EPA (C20:5), docosapentaenoic acid (DPA, C22:5) and DHA (C22:6) as ω-3 PUFA. These values were also reported as total FA contents (total SFA, total MUFA, total PUFA, total ω-6, total ω-3).

Lipid indexes were calculated as follows: ω-6-to-ω-3 ratio (ω-6/ω-3), PUFA balance [ω-3/(ω-3 + ω-6)], SFA-to-MUFA ratio (SFA/MUFA), unsaturation index (UI = MUFA total x 1 + C18:2 x 2 + C20:3 x 3 + C20:4 x 4 + C20:5 x 5 + C22:6 x 6) and peroxidation index (PI = MUFA total x 0.025 + C18:2 x 1 + C20:3 x 2 + C20:4 x 4 + C20:5 x 6 + C22:6 x 8) [23, 27]. The elongase-6 activity index (EI = C18:0/C16:0), Δ-9 desaturase activity index (Δ9DI = 9c, C18:1/C18:0), Δ-6 desaturase activity index (Δ6DI = C20:3/C18:2), and Δ-5 desaturase index (Δ5DI = C20:4/C20:3) were calculated as precursors to FA ratios, as already described in dogs [23].

## Statistical analysis

The computer software GraphPad Prism version 6.01 (GraphPad Software San Diego, CA) was utilized to perform the statistical analysis. All data were evaluated using a standard descriptive statistic and reported as mean ± SD or as median and interquartile range, based on

their distribution. Normality was checked using the D'Agostino Pearson test. For comparison between two groups, the unpaired t-test or Mann-Whitney test was used, while for comparison among more than two groups, ANOVA or Kruskal-Wallis test and post-hoc tests (Student-Newman-Keuls test or Dunn test) were employed. Correlations between FA percentages, homeostasis indexes or enzyme activity indexes with FECAI and BCS were evaluated by Spearman or Pearson correlation analysis depending on their distribution. The threshold of statistical significance was set at $p < 0.05$.

Principal coordinate analysis and hierarchical clustering heatmaps were generated using R Studio and Metaboanalyst 5.0, respectively.

## Results

### Study population

Eighty-three cats, of which 41 were diagnosed with FCE and 43 were HC, underwent FA-based RBC membrane lipidome analysis (Tables 2 and 3).

Among the FCE group, 17 cats responded positively to the dietary trial and were diagnosed with FRE. Of the 24 cats that failed to respond positively to at least two dietary trials, 15 were diagnosed with IBD, and 9 with LGITL, based on biopsy and histopathological evaluations.

Among the FCE cats, the FCEAI was available in 34, with a mean value of 7.6 ± 3.9. Folates and cobalamin levels were available in 31 cats; 14 had cobalamin below the RI (290–1500 ng/L), 17 had cobalamin levels within the RI, while no cats were hypercobalaminemic 9 had folate below the RI, 16 had folate levels within the reference interval, and 6 had folate levels above the reference interval. The clinical findings observed in FCE are summarized in Table 4.

### Fatty acid-based erythrocyte membrane lipidome analysis

Median values with interquartile ranges of the single FAs, total FA contents (total SFAs, total MUFA, and total PUFA), homeostasis indexes (SFA/MUFA, ω-6/ω-3, UI, PI, and PUFA balance) and enzyme activity indexes (EI, Δ9DI, Δ6DI, Δ5DI) of FCE cats are reported in Table 5. Graphical representations of variations in fatty acids among the three groups of cats, is presented as principal coordinate analysis (Fig 1) and hierarchical clustering heatmaps (Fig 2).

Compared to HC, in the RBC membranes of FCE cats higher values of DPA (p = 0.0002) and DHA (p = 0.0246) were observed, with an overall increase of RBC content of ω-3 PUFA (p = 0.006), and lower values of RBC content of ω-6 PUFA linoleic acid (p = 0.0026). Additionally, the membrane homeostasis index patterns in FCE cats showed an increased PUFA balance (p = 0.0019) and a decreased ω-6/ω-3 ratio (p = 0.0019); also, Δ6DI resulted increased in FCE cats (p = 0.0151) (Figs 3 and 4).

Regarding the diagnosis, no differences were observed in FA-based RBC membrane lipidome among FRE, IBD, and LGITL cats (S1 Table).

In comparison with FCE cats with normal levels of serum cobalamin (n = 17), hypocobalaminemic FCE cats (n = 14) had decreased values of total PUFA (p = 0.0467), SFA/MUFA ratio (p = 0.0005), and the value of PI (p = 0.0467), while the values of oleic acid (p = 0.0030) and vaccenic acid (p = 0.0155), total MUFA (p = 0.0013), Δ9DI (p = 0.0008) and Δ5DI (p = 0.003) were increased. No differences in RBC membrane fatty acids levels, lipid indexes and enzyme indexes were observed among hypofolatemic, normofolatemic, and hyperfolatemic cats.

Linoleic acid (r = 0.43, 95% CI [0.11; 0.67], n = 34, p = 0.01), DPA (r = 0.47, 95% CI [0.16; 0.42], n = 34, p = 0.004), ω-6 PUFA (r = 0.38, 95% CI [0.04; 0.63], n = 34, p = 0.03) and total

**Table 2. Breed, sex, age, body weight and diagnosis of the recruited cats with feline chronic enteropathy (n = 41).**

|  | Breed | Sex | Age (month) | Body weight | Diagnosis |
|---|---|---|---|---|---|
| 1 | Domestic Shorthair | Mn | 20 | 4.6 | IBD |
| 2 | Domestic Shorthair | Fs | 144 | 3 | FRE |
| 3 | Domestic Shorthair | Fs | 132 | 3 | IBD |
| 4 | Domestic Shorthair | Fs | 108 | 3 | LGITL |
| 5 | Domestic Shorthair | Mn | 48 | 4 | IBD |
| 6 | Domestic Shorthair | M | 34 | 3.6 | FRE |
| 7 | Domestic Shorthair | Fs | 60 | 2.7 | FRE |
| 8 | Domestic Shorthair | F | 120 | 3 | LGITL |
| 9 | Domestic Shorthair | Mn | 120 | 7.3 | FRE |
| 10 | Domestic Shorthair | Mn | 18 | 5.2 | FRE |
| 11 | Siamese | Fs | 48 | 3.9 | FRE |
| 12 | Domestic Shorthair | Fs | 8 | 2 | FRE |
| 13 | Domestic Shorthair | Mn | 108 | 3.65 | LGITL |
| 14 | Domestic Shorthair | Mn | 18 | 2.4 | FRE |
| 15 | Domestic Shorthair | Mn | 84 | 6.9 | FRE |
| 16 | Domestic Shorthair | Fs | 24 | 2.25 | FRE |
| 17 | Domestic Shorthair | Fs | 120 | 4.9 | FRE |
| 18 | Domestic Shorthair | Fs | 120 | 6.7 | FRE |
| 19 | Domestic Shorthair | Mn | 120 | 3.6 | LGITL |
| 20 | Domestic Shorthair | M | 156 | 4 | LGITL |
| 21 | Domestic Shorthair | Mn | 147 | 3.3 | LGITL |
| 22 | Domestic Shorthair | Fs | 182 | 4.1 | LGITL |
| 23 | Domestic Shorthair | Fs | 108 | 3.3 | FRE |
| 24 | Domestic Shorthair | Fs | 48 | 3 | IBD |
| 25 | Domestic Shorthair | Fs | 96 | 6.3 | IBD |
| 26 | Domestic Shorthair | Mn | 155 | 3 | LGITL |
| 27 | Domestic Shorthair | M | 138 | 5.4 | IBD |
| 28 | Domestic Shorthair | Mn | 128 | 5.3 | IBD |
| 29 | British Shorthair | Mn | 36 | 3.9 | IBD |
| 30 | Maine Coon | Mn | 60 | 5 | IBD |
| 31 | Abyssinian | Mn | 117 | 4.1 | FRE |
| 32 | Domestic Shorthair | Fs | 96 | 4.1 | IBD |
| 33 | Domestic Shorthair | Mn | 124 | 3 | IBD |
| 34 | Domestic Shorthair | Fs | 120 | 3.2 | IBD |
| 35 | Carthusian | Fs | 120 | 3.2 | IBD |
| 36 | Domestic Shorthair | Mn | 120 | 7.2 | FRE |
| 37 | Domestic Shorthair | Fs | 156 | 3.4 | LGITL |
| 38 | Domestic Shorthair | Mn | 58 | 4.3 | FRE |
| 39 | Main coon | F | 132 | 4.2 | IBD |
| 40 | Domestic Shorthair | M | 5 | 7 | FRE |
| 41 | Devon rex | M | 68 | 2 | IBD |

Male; F: Female; Mn: Neutered male; Fs: Spayed female; FRE: Food-responsive enteropathy; IBD: inflammatory bowel disease; LGITL: low-grade intestinal T-cell lymphoma.

Table 3. Breed, sex, age, body weight and diagnosis of the recruited healthy cats (n = 43).

| | Breed | Sex | Age (month) | Body weight |
|---|---|---|---|---|
| 1 | Domestic Shorthair | Fs | 55 | 4 |
| 2 | Domestic Shorthair | Fs | 69 | 5 |
| 3 | Domestic Shorthair | Fs | 136 | 2.5 |
| 4 | Domestic Shorthair | F | 19 | 3.6 |
| 5 | Domestic Shorthair | Fs | 83 | 3.3 |
| 6 | Domestic Shorthair | Fs | 9 | 3 |
| 7 | Domestic Shorthair | Mn | 36 | 4.2 |
| 8 | Siamese | Mn | 23 | 4.5 |
| 9 | Domestic Shorthair | Fs | 132 | 5.7 |
| 10 | Domestic Shorthair | M | 60 | 5 |
| 11 | Domestic Shorthair | Fs | 60 | 3 |
| 12 | Domestic Shorthair | Fs | 10 | 3.5 |
| 13 | Domestic Shorthair | Mn | 36 | 4.5 |
| 14 | Domestic Shorthair | Mn | 65 | 5.8 |
| 15 | Domestic Shorthair | Fs | 7 | 3.2 |
| 16 | Domestic Shorthair | Mn | 5 | 4.7 |
| 17 | Domestic Shorthair | Fs | 72 | 3 |
| 18 | Domestic Shorthair | Mn | 24 | 4 |
| 19 | Domestic Shorthair | Fs | 24 | 4.5 |
| 20 | Domestic Shorthair | M | 36 | 4.25 |
| 21 | Domestic Shorthair | Fs | 24 | 2.3 |
| 22 | Domestic Shorthair | Mn | 24 | 3.8 |
| 23 | Domestic Shorthair | F | 12 | 2.1 |
| 24 | Domestic Shorthair | F | 24 | 2.2 |
| 25 | Domestic Shorthair | Mn | 12 | 3.3 |
| 26 | Domestic Shorthair | F | 12 | 1.8 |
| 27 | Domestic Shorthair | F | 37 | 3 |
| 28 | Domestic Shorthair | F | 13 | 2.4 |
| 29 | Domestic Shorthair | Fs | 33 | 2.3 |
| 30 | Domestic Shorthair | F | 45 | 2.5 |
| 31 | Domestic Shorthair | M | 57 | 4.2 |
| 32 | Domestic Shorthair | M | 33 | 2.8 |
| 33 | Domestic Shorthair | Mn | 36 | 4.2 |
| 34 | Domestic Shorthair | Mn | 60 | 5 |
| 35 | Domestic Shorthair | Fs | 120 | 3.4 |
| 36 | Domestic Shorthair | Mn | 132 | 3.8 |
| 37 | Domestic Shorthair | Mn | 108 | 4.1 |
| 38 | Domestic Shorthair | Fs | 163 | 3.8 |
| 39 | Domestic Shorthair | Mn | 163 | 5.2 |
| 40 | Domestic Shorthair | Fs | 118 | 4.8 |
| 41 | Domestic Shorthair | Mn | 67 | 6.1 |
| 42 | Domestic Shorthair | M | 9 | 4.2 |
| 43 | Domestic Shorthair | Mn | 19 | 4.7 |

M: Male; F: Female; Mn: Neutered male; Fs: Spayed female

**Table 4. Clinical and clinicopathological findings observed in cats with chronic enteropathy.**

| Variables | n (%) |
|---|---|
| Vomiting | 22/41 (53.6%) |
| Diarrhea | 22/41 (53.6%) |
| Weight loss | 22/41 (53.6%) |
| Decreased serum cobalamin (< 290 ng/l) | 14/31 (45.1%) |
| Decreased serum folate (< 9.7 µg/l) | 9/31 (29%) |
| Decreased appetite | 10/41 (24.3%) |
| Increased serum folate (> 21.6 µg/l) | 6/31 (19.3%) |
| Decreased attitude/activity | 2/41 (4.88%) |

PUFA (r = 0.42, 95% CI [0.09; 0.66], n = 34, p = 0.01) were positively correlated with the FCEAI, while a negative correlation with FCEAI was observed for palmitic (r = -0.40, 95% CI [-0.65; -0.07], n = 34, p = 0.01), oleic acids (r = -0.30, 95% CI [-0.63; -0.04], n = 34, p = 0.02) and SFA (r = -0.35, 95% CI [-0.61; -0.01], n = 34, p = 0.04).

**Table 5. Median values with interquartile ranges in brackets of the single FAs, total FA contents of red blood cells membranes (total SFA, total MUFA, and total PUFA), homeostasis indexes (SFA/MUFA, ω-6/ω-3, UI, PI, and PUFA balance) and enzyme activity indexes (EI, Δ9DI, Δ6DI, Δ5DI) for both healthy cats and cats with FCE.** Statistically significant *p*-values (*p* < 0.05) are highlighted in bold.

| Variable | Healthy Cats (n = 43) Median value (IQR) | FCE (n = 41) Median value (IQR) | p value |
|---|---|---|---|
| C16:0—Palmitic Acid | 18.97 (16.1–22.3) | 19.1(17.4–22.6) | 0.4760 |
| C16:1—Palmitoleic Acid | 0.17 (0.11–0.22) | 0.14 (0.10–0.20) | 0.3234 |
| C18:0—Stearic Acid | 20.4 (23–24.6) | 22.4 (20.7–24.3) | 0.6498 |
| 9c,C18:1—Oleic Acid | 8.32 (9.23–10) | 9.94 (0.05–12.1) | 0.0892 |
| 11c,C18:1—Vaccenic Acid | 1.45 (1.73–1.96) | 1.82 (1.50–2.27) | 0.1204 |
| LA ω-6—C18:2—Linoleic Acid | 21.0 (23.2–25.4) | 20.6 (17.8–23.5) | **0.0026** |
| DGLA ω-6—C20:3 Dihomogammalinolenic Acid | 0.54 (0.75–0.92) | 0.80 (0.64–1.06) | 0.2042 |
| ARA ω-6—C20:4—Arachidonic Acid | 19.3 (16.1–23.3) | 19.9 (14.8–23.7) | 0.9840 |
| EPA ω-3—C20:5—Eicosapentaenoic Acid | 0.90 (0.64–1.50) | 1.44 (0.66–2.58) | 0.0555 |
| DPA ω-3—C22:5—Docosapentaenoic Acid | 0.50 (0.34–0.67) | 0.74 (0.51–0.74) | **0.0002** |
| DHA ω-3—C22:6—Docosahexaenoic Acid | 0.90 (0.69-1-31) | 1.36 (0.62–1.98) | **0.0246** |
| Total SFA | 42.5 (36.5–46.2) | 41.4 (39.8–45.4) | 0.9947 |
| Total MUFA | 11.3 (10.28–11.9) | 11.9 (9.67–14.5) | 0.0939 |
| Total PUFA | 46.55 (41.5–51.5) | 45.5 (42.1–49.3) | 0.4942 |
| ω-6 PUFA | 43.6 (39.0–48.5) | 41.1 (36.7–45) | 0.0552 |
| ω-3 PUFA | 2.35 (1.87–2.87) | 3.48 (2.29–5.36) | **0.0026** |
| ω-6/ω-3 ratio | 19.7 (14.6–22.1) | 13.0 (7.82–19.2) | **0.0019** |
| SFA/MUFA | 3.79 (3.43–4.15) | 3.64 (2.85–4.21) | 0.2687 |
| PUFA balance | 4.82 (4.32–6.38) | 7.12 (4.93–11.35) | **0.0019** |
| UI | 151.9 (136.1–167.5) | 158.6 (139.8–170) | 0.4651 |
| PI | 122.2 (105.4–139.1) | 131.8 (110.4–147.1) | 0.1055 |
| Elongase-6 activity | 1.22 (0.58–1.37) | 1.16 (1.01–1.28) | 0.1428 |
| Delta-9 desaturase | 0.40 (0.35–0.43) | 0.42 (0.34–0.52) | 0.1135 |
| Delta-6 desaturase | 0.32 (0.02–0.04) | 0.03 (0.02–0.060) | 0.0151 |
| Delta-5 desaturase | 23.9 (20.1–32.6) | 23.13 (17.1–30.9) | 0.4022 |

IQR: interquartile range; SFA: Saturated Fatty Acids; MUFA: Monounsaturated Fatty Acids; PUFA: Polyunsaturated Fatty Acids; UI: Unsaturation index; PI: Peroxidation index.

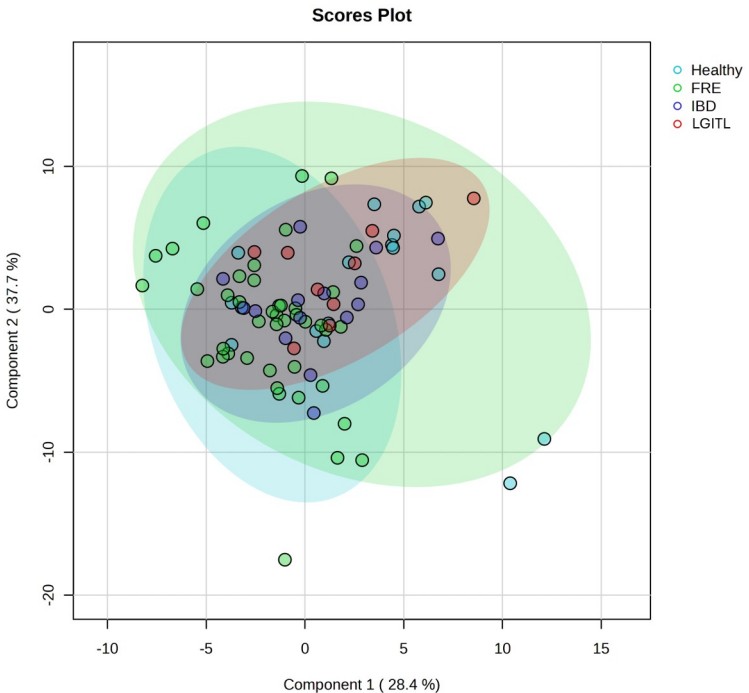

**Fig 1. Plots of the scores for the two principal components (variance explained: 28.4% and 37.7%, respectively) of healthy cats and cats with food-responsive enteropathy (FRE), inflammatory bowel disease (IBD) and low-grade intestinal T-cell lymphoma (LGITL).**

The BCS was found positively correlated with DHA (r = 0.33, 95% CI [0.01; 0.58], n = 41, p = 0.03), total ω-3 (r = -0.31, 95% CI [0.01; 0.56], n = 41, p = 0.04) and PUFA balance (r = 0.32, 95% CI [0.01; 0.57], n = 41, p = 0.04), while a negative correlation with ω-6/ω-3 (r = -0.32, 95% CI [-0.57; -0.001], n = 41, p = 0.04) (p = 0.0442, r = − 0.32) was observed.

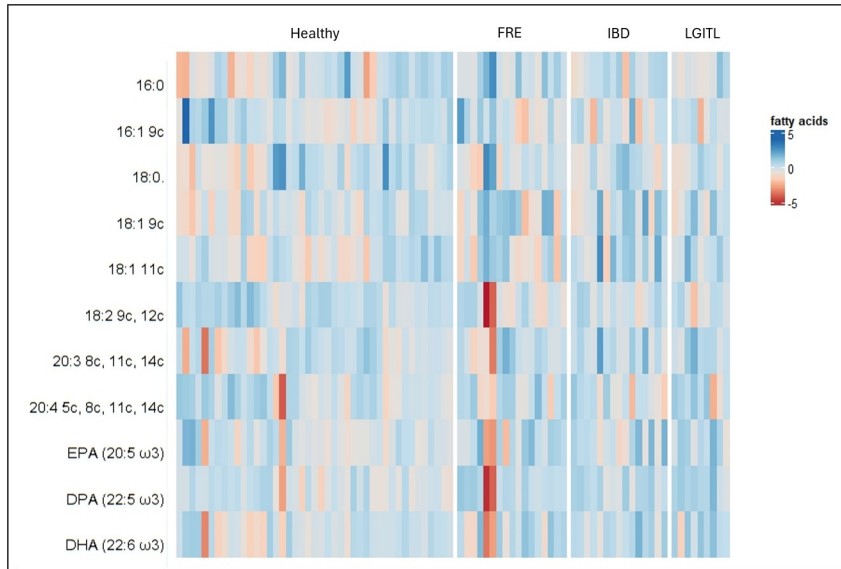

**Fig 2. Heatmap of the fatty acid of red blood cells membranes.** Each row represents the intensity of one fatty acid; each column represents one sample. The higher the concentration of a fatty acid, the more intensely red shows. The lower the signal intensity of a fatty acid, the more intensely blue the metabolite shows in the heatmap.

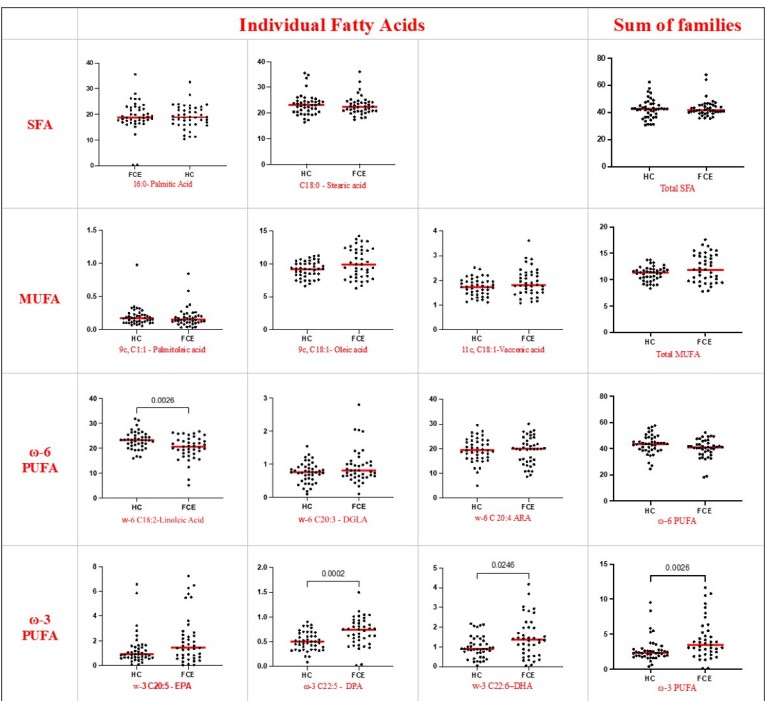

**Fig 3. Red blood cell membranes concentration of fatty acids in healthy cats (n = 43) and cats with chronic enteropathy (n = 41).** Red lines represent the median. Statistically significant p-values (p-value <0.05) are showed in thew graphs. SFA: Saturated Fatty Acids; MUFA: Monounsaturated Fatty Acids; PUFA: Polyunsaturated Fatty Acids.

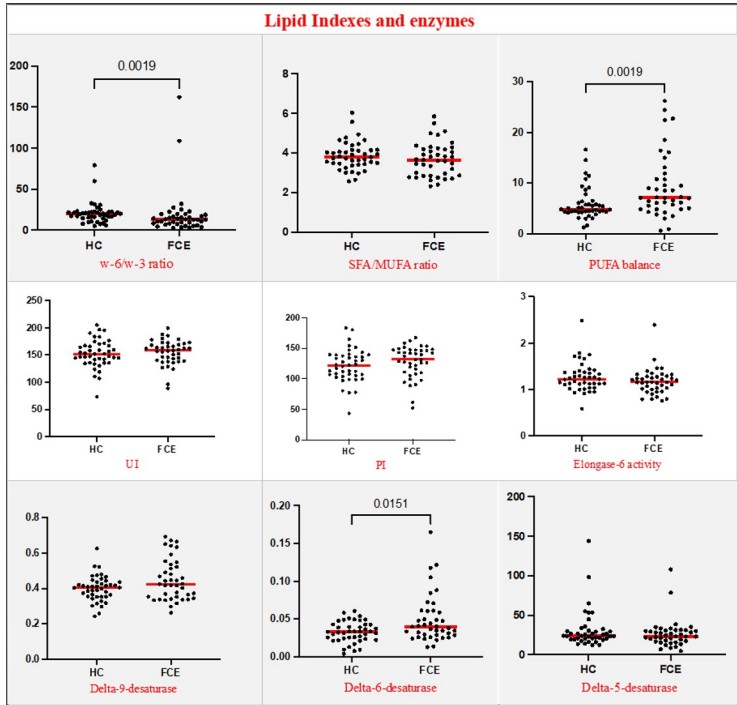

**Fig 4. Fatty acids indexes and enzymes indexes in healthy cats (n = 43) and cats with chronic enteropathy (n = 41).** Red lines represent the median. Statistically significant p-values (p-value <0.05) are showed in the graphs. SFA: Saturated Fatty Acids; MUFA: Monounsaturated Fatty Acids; PUFA: Polyunsaturated Fatty Acids; UI: Unsaturation index; PI: Peroxidation index.

## Discussion

The results of the present study revealed distinct differences in the FA composition of RBC membranes between FCE cats and the HC group. The most consistent differences observed in FCE compared to healthy ones were related to PUFA metabolism. In FCE, increased DPA and DHA levels and, consequently, total ω-3 PUFA, were observed, alongside a decrease in LA levels. Accordingly, increased PUFA balance and Δ6DI and reduced ω-6/ω-3 ratio was also recorded. These results are consistent with those observed in dogs with CE [23], supporting the hypothesis that the lipidic imbalance observed in CE is very similar in the two species, despite clear metabolic differences. Indeed, cats possess limited Δ6 desaturase activity and alternative enzymatic pathways mediated by Δ-5 and Δ-8 desaturase to produce AA have been evocated [28]. For this reason, cats have a greater requirement for dietary AA, EPA, and DHA, than dogs, especially for pregnant and lactating cats for offspring's brain and nervous system development. The FEDIAF Nutritional Guidelines for Complete and Complementary Pet Food for Cats and Dogs recommend the following FAs levels for complete cat food Unit per 1000 kcal of metabolizable energy (ME), based on ME Requirements of 75 Kcal/Kg$^{0.67}$: 1.67 grams of LA and 20 mg of AA for adult cats, and 1.38 grams of LA, 50 mg of AA, 0.05 grams of α-linolenic acid and 0.03 grams of EPA+DHA [29]. In contrast, dogs, with their active Δ6 desaturase, efficiently convert linoleic acid to arachidonic acid, and can synthesize EPA and DHA from α-linolenic acid. Additionally, dogs exhibit high stearoyl-CoA desaturase activity, aiding in the conversion of palmitic acid to unsaturated fatty acids [28].

Regarding moisture content, there is a lack of available data concerning the bioavailability of FAs in cats consuming heat-treated canned food. However, recent research indicates that moisture levels may influence the production of short-chain FAs, potentially attributable to augmented substrate reaching the colon for fermentation, particularly observed in cats consuming a dry diet [30].

Interestingly, as already observed in dogs [23], the total ω-6 and AA levels did not differ in the RBC membranes of FCE and HC, further recalling the complex biochemical relationship between ω-3 and ω-6 pathways in pathological conditions. Indeed, the balance between ω-6 and ω-3 PUFA is critical for regulating the signaling outcome in chronic enteropathy [31].

While few studies in cats evaluated the effects of diet on different blood compartments (i.e. serum, plasma, RBC membrane) and tissue [32, 33], to the authors' knowledge, this is the first study investigating the FA cohort of the RBC membrane lipidome in cats affected by CE. A cluster of 11 FAs representative of each family, including SFA, MUFA, and PUFA, was selected based on their structural and functional roles, and their relative abundance in the RBC membrane, since it belongs to the most representative FAs of the membranes, accounting for approximately 97% of the total content.

The analysis of FA levels in plasma or serum has been extensively conducted as it provides insights into short-term dietary fat consumption [34]. Nonetheless, examining the lipid composition in RBC membranes offers distinct advantages over plasma analysis. This is primarily because erythrocytes have a longer lifespan (i.e. approximatively 120 days in humans and dogs, 65–75 days in cats) in the bloodstream making them a more reliable indicator of long-term dietary FA intake and overall tissue conditions [34]. Additionally, it's worth noting that erythrocytes tend to maintain a more consistent and stable FA composition compared to the levels found in plasma, as it is generally believed that these cells keep their FA distribution throughout their life. [35].

Moreover, FA-based RBC membrane lipidome may provide an estimation of the clinical and nutritional status of the patients, and of the metabolic pathways in which they are mostly involved [36]. In particular, the SFA and MUFA levels provide information on the

liponeogenesis and Δ-9 desaturase activity, while essential PUFA are indicative of the nutritional status, and their metabolites indicate the efficiency of further steps of elongation, Δ-6 and Δ-5 desaturations [37–40].

The result of the present study suggested a dysregulation in the balance between ω-3 and ω-6 in cats with CE. The role of FAs in inflammation is central to the understanding of the pathophysiology of various conditions, including chronic inflammatory gastrointestinal disorders. In this context ω-6 PUFA, specifically LA, are recognized as key precursors for the biosynthesis of AA and a multitude of inflammatory mediators, including prostaglandins and leukotrienes, which are well-documented for their pro-inflammatory properties [41]. On the contrary, ω-3 PUFA are acknowledged for their anti-inflammatory attributes, largely achieved by competing with ω-6 PUFA for enzymatic conversion [42]. Notably, ω-3 PUFA also serve as precursors for a distinct class of eicosanoids, including leukotriene B5, which possesses either negligible or anti-inflammatory properties, and anti-inflammatory lipid mediators known as pro-resolving mediators [43, 44].

The ω-3 PUFA, EPA and DHA, are known to exert their anti-inflammatory effects in the enterocytes through the partial replacement of AA in cellular phospholipids, the inhibition of several inflammatory signaling pathways, the activation of peroxisome proliferator-activated receptor (PPAR)-γ and G-protein coupled receptor, which subsequently inhibit the action of the proinflammatory transcription factor nuclear kappa-light-chain-enhancer of activated B cells (NF-κB) [45].

Given that both AA and ω-3 PUFA share the same enzymes, it has been observed that the incorporation of ω-3 PUFA into the cell membrane of erythrocyte occurs at the expense of AA [46] with ω-3 being preferentially utilized [41]. The data collected from this FCE study in cats and the previous study in CE dogs [23] allowed us to hypothesize that the inflammatory process, through upregulation of phospholipase A2 expression, may lead to a high rate of detachment from cell membranes of AA and a subsequent imbalance of substrate competition and the accumulation of ω-3 in membranes. These results seem in agreement with a recent study by Marsilio and coll. [47] in which an increase of fecal arachidonate was found in cats, suggesting mucosal upregulation during inflammation and subsequent leakage into the lumen. Similarly, dogs with FRE tended to have greater AA in feces when compared to the control group, suggesting excessive membrane destruction in sick dogs or to greater production of this fatty acid to repair cellular damage at the intestinal level [48].

The ω-6/ω-3 ratio, which is also referred to as the inflammatory risk index in humans [49], can be managed through dietary management to reduce intestinal inflammation in humans and animals, and the PUFA balance is a usefulness index for assessing the overall effects of PUFA on membranes [50]. This index could aid in determining the role of nutraceutical intervention and, specifically, the appropriateness of ω-3 supplementation in feline CE. Recently, researchers have investigated the ω-3 index in both dogs and cats, which reflects the content of EPA and DHA in erythrocytes expressed as a percentage of total RBC membrane FAs [32]. These indexes can aid in estimating the appropriate amount of ω-3 supplementation required in cats affected by CE.

Other than EPA and DHA, in this study, the ω-3 PUFA DPA was considered in the FA lipidomic profile of cats. DPA (22:5) is the precursor of the DHA, and it derives from the EPA (20:5) by the elongase enzymatic activity [51]. DHA levels in RBC membranes of dogs were negligible [23, 52]. On the other hand, adult cats can produce DPA in liver and plasma [28] representing approximately 0.5% of the total FA content of RBC membranes of HC in the present study, with EPA and DHA approximately 0.9% each, and interestingly DPA was significantly increased in FCE. Like EPA and DHA, DPA is a substrate for the synthesis of specialized pro-resolving mediators with biological activity (i.e., resolvins and protectins), suggesting

a role in the resolution of inflammation [53, 54]. Moreover, the rapid conversion between EPA and DPA indicates the possibility that DPA can be a potential storage form for EPA [54], suggesting an impact of DPA on various aspects of chronic inflammation in cats.

Similarly to the previous findings in the study on dogs [23], the composition of RBC membranes in terms of FA percentages in cats was not capable of discriminating among the different forms of chronic enteropathy (FRE, IBD, LGITL). This outcome is not surprising, as currently there is no single diagnostic criterion or established biomarker available that can reliably differentiate IBD from LGITL of cats [4, 55–58]. Furthermore, it is noteworthy that both conditions often coexist within the same individual, which poses a challenge in distinguishing the various forms of FCE [5]. Another plausible explanation for this difficulty could be that the chronic inflammatory process characterizes all three forms independently of their specific triggering cause [5].

Hypocobalaminemia has been frequently reported in FCE and has been demonstrated to have a correlation with the severity of clinical signs in cats [59, 60]. Along with folate, cobalamin serves as an indicator of jejunal and ileal malabsorption, respectively, providing valuable information about the location of the disease [61]. In the present study hypocobalaminemic cats exhibited an increase in oleic acid, vaccenic acid, and total MUFA. Subsequently, elevated activity of Δ-9 desaturase was observed, which may account for the substantial accumulation of MUFA in these cases. These findings support the hypothesis that this enzyme may play a crucial role in the pathogenesis of FCE. Similarly, in dogs, a negative correlation was observed between cobalaminemia and vaccenic acid levels in RBC membranes, with an increase of vaccenic acid in dogs with CE that did not respond to therapy [23].

Among FCE cats, an increase in FCEAI was found to be associated with higher levels of LA, DPA, ω-6 PUFA, and total PUFA. This higher rate of PUFA incorporation impacts the distribution of FAs in RBC membranes, leading to lower levels of SFA at higher clinical activity index scores. We can also speculate that higher PUFA levels in RBC membranes may reflect a greater susceptibility to oxidative damage, which is known to occur during inflammation [62]. This finding supports the hypothesis that ω-3 may be upregulated in both mild and severe diseases, and their consumption may increase as the disease activity becomes more severe [63] as already suggested in dogs with CE [23].

While in CE dogs the FA-based RBC membrane lipidome did not appear to be influenced by the BCS [23], in FCE, lower values of BCS were associated with reduced levels of DHA and total ω-3 with subsequent significant increase of ω-6/ω-3. In these subjects, factors such as malabsorption, oxidative decomposition of PUFA, low desaturation activity, and rapid turnover of RBCs may all contribute to the lower DHA levels.

This study represents the initial examination of the RBC membrane FA profile in FCE. However, it is important to acknowledge several limitations. Firstly, the sample size was restricted, which may have reduced the statistical power of the analyses. Moreover, the absence of a standardized diet before enrolling the cats could potentially impact the interpretation of the results, and implementing a standardized, high-quality diet with known percentages of FA content could address this limitation. Finally, conducting follow-up sampling could have provided valuable insight into treatment response.

The study suggests that future investigations on the FA lipidomic profile of RBC membranes in FCE could offer potential benefits in monitoring treatment response, personalized nutraceutical approaches, and identifying markers of disease relapse. Prospective studies under controlled conditions and with standardized diets would be necessary to further explore the effects of diet and treatment on lipidomic changes. Recent advancements in lipidomics provide opportunities for comprehensive lipidome mapping and monitoring in humans and animals, including cats, to understand the impact of chronic enteropathy on animal health.

## Conclusion

In summary, the present study highlights significant FA changes in RBC membrane in FCE compared to HC with the most notable differences were observed in the ω-3 PUFA (DPA, DHA, total ω-3 PUFA). A previous study suggested that changes in RBC membranes, in particular ω-3 PUFA, seem to reflect changes in gastrointestinal tissue [64], suggesting that FA-based RBC membrane lipidome approach may represent a valid and less invasive test for investigating the health of the gastrointestinal tract.

The applicability FA-based RBC as an indicator for specific health outcomes, including feline chronic enteropathy, requires further investigation to be confirmed.

## Supporting information

**S1 Table. Median values with interquartile ranges in brackets of the single FAs, total FA contents of red blood cells membranes (total SFA, total MUFA, and total PUFA), homeostasis indexes (SFA/MUFA, ω-6/ω-3, UI, PI, and PUFA balance) and enzyme activity indexes (EI, Δ9DI, Δ6DI, Δ5DI) in the different groups of FCE cats: Food-responsive enteropathy (FRE), inflammatory bowel disease (IBD) and low-grade intestinal T-cell lymphoma (LGITL).** SFA: Saturated Fatty Acids; MUFA: Monounsaturated Fatty Acids; PUFA: Polyunsaturated Fatty Acids; UI: Unsaturation index; PI: Peroxidation index.
(DOCX)

## Author Contributions

**Conceptualization:** Paolo Emidio Crisi, Alessandro Gramenzi, Paraskevi Prasinou, Anna Sansone, Carla Ferreri, Andrea Boari.

**Data curation:** Maria Veronica Giordano, Anna Sansone, Andrea Boari.

**Formal analysis:** Paolo Emidio Crisi, Maria Veronica Giordano, Paraskevi Prasinou, Anna Sansone.

**Investigation:** Paolo Emidio Crisi, Maria Veronica Giordano, Alessia Luciani, Paraskevi Prasinou, Anna Sansone, Valentina Rinaldi, Carla Ferreri.

**Methodology:** Alessandro Gramenzi, Carla Ferreri, Andrea Boari.

**Supervision:** Alessandro Gramenzi, Carla Ferreri, Andrea Boari.

**Validation:** Andrea Boari.

**Writing – original draft:** Paolo Emidio Crisi, Maria Veronica Giordano.

**Writing – review & editing:** Alessia Luciani, Alessandro Gramenzi, Anna Sansone, Valentina Rinaldi, Carla Ferreri, Andrea Boari.

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
