## [Decision Letter · Decision Letter 0]

10 Mar 2024

PONE-D-24-03902Evaluation of the fatty acid-based erythrocyte membrane lipidome in cats with Food Responsive Enteropathy, Inflammatory Bowel Disease and Low-Grade Intestinal T-Cell LymphomaPLOS ONE

Dear Dr. Crisi,

Thank you for submitting your manuscript to PLOS ONE. After careful consideration, we feel that it has merit but does not fully meet PLOS ONE’s publication criteria as it currently stands. Therefore, we invite you to submit a revised version of the manuscript that addresses the points raised during the review process.

We look forward to receiving your revised manuscript.

Kind regards,

Juan J Loor

Academic Editor

PLOS ONE

Journal Requirements:

2. We noted in your submission details that a portion of your manuscript may have been presented or published elsewhere. [A related paper, intended to assess the impact of age, body weight, sex, and lifestyle on the fatty acid-based erythrocyte membrane lipidome of healthy cats, is currently under review by the journal Veterinary Medicine International. The cohort of healthy cats examined in this related paper is the same being considered in the herein submitted manuscript] Please clarify whether this [conference proceeding or publication] was peer-reviewed and formally published. If this work was previously peer-reviewed and published, in the cover letter please provide the reason that this work does not constitute dual publication and should be included in the current manuscript.

3. We notice that your supplementary [S1 and S2 Fig.] are included in the manuscript file. Please remove them and upload them with the file type 'Supporting Information'. Please ensure that each Supporting Information file has a legend listed in the manuscript after the references list.

Reviewers' comments:

Reviewer's Responses to Questions

**Comments to the Author**

1. Is the manuscript technically sound, and do the data support the conclusions?

Reviewer #1: Yes

2. Has the statistical analysis been performed appropriately and rigorously? 

Reviewer #1: No

3. Have the authors made all data underlying the findings in their manuscript fully available?

Reviewer #1: Yes

4. Is the manuscript presented in an intelligible fashion and written in standard English?

Reviewer #1: No

5. Review Comments to the Author

Reviewer #1: I would like to express my gratitude to the authors for sharing their fascinating study. The paper delves into the fatty acid profiles of the RBC membrane in both healthy cats and those with chronic enteropathies, with a thorough and well-articulated description of the study populations and methods. The introduction and discussion are highly informative, providing valuable insights into the topic. However, the results may benefit from some refinement, and the references require meticulous editing. Any typographical or grammatical errors should be corrected.

General comments:

• Please reconcile CE or FCE throughout the manuscript

• Please reconcile PUFA w-3 or w-3 PUFA

Detailed comments:

Line 1: Title: mixed case or sentence case?

Abstract

The word count of the abstract exceeds the limit. Please carefully read the author's guidelines.

Line 30, Gas chromatography “with which detection device (UV, MS, or others)” was used to quantitatively analyze a cluster of 11 “FAs”,

Line 31, “Including lipid homeostasis and enzyme activity indexes” This part is confusing. Those are not directly measured by the GC, but a calculation based on the results. Make it clear.

Line 34, “PUFA” should not be abbreviated without the full name if only appears once.

Line 37, “subgroups” of CE. No paragraphs in the abstract.

Line 38, In “humans” and dogs

Introduction

Line 80: UC and CD first appear.

Line 84: Check the abbreviation and provide their full names if first appear.

Materials and methods

Line 107: be specific, which serum biochemistry profile

Line 108: feline “serum” pancreatic lipase “i…”

Lines 128-132: need more polish.

Line 142: FAME?

Lines 145-149: please explain the information in the parentheses.

Line 150: explains which are SFA, MUFA, PUFA, w-6 and w-3. A table summarizing the information would be appreciated.

Line 162-163: Did you test the normality of your data? Which tests? Did you correct the p-values for multiple comparisons?

Results

Line 171: Basic information is lacking. Signalment for both groups. Move the S1 and S2 tables to the main text.

Line 174: Be specific “positively” what does it mean? Decreased of FCEAI? No clinical signs?

Line 191: PCA and heatmap might be good ways to visualize this dataset with multiple variables. Please add them for readers to better appreciate the changes.

Line 197: Are these P values corrected for multiple comparisons? Please add adjusted P-values or Q-values. What are the units of these variables? “IQR” is a better abbreviation for interquartile range.

Lines 201-207: Please add and make it clear that the FAs are from “RBC membranes” in the context and captions of tables and figures. For example, concentrations of DPA in RBC membranes. FAs can be measured in other biological matrixes as well. Some representative figures would help readers to appreciate the differences between groups. The quality of the supplementary figures is poor. I am not sure if it’s due to the end of the submission portal or from the authors.

Line 206: Table 3 can be in the supplementary data or have some representative figures of some results.

Line 217: Hypocobalaminemic cats (n=?), the changes were compared to which population?

Line 220: No difference of what?

Lines 222-225: Please add the 95% confidence interval and the numbers of observations of each statistic. For example, xyz acid (r = 0.25, 95% CI [0.18-0.33], n = 65, p = 0.002)

Discussion

While this journal does not impose any word limit, it is recommended to consider editing/removing certain portions of both mice and human studies. This would ensure that the content is concise and impactful, leading to better engagement and understanding among readers in veterinary medicine.

Line 238: Kindly explain how metabolic differences between the two species.

Line 243: Do you know if certain fatty acids are essential to add to cat’s diet according to NRC, AFFCO, or other authorities? What are the most common fatty acids found in their diet? Any differences between extruded diet and canned food?

Line 245: cats “with CE”. Please reconcile throughout the whole manuscript.

Line 250-251: missing reference.

Line 253: be specific, how long is the lifespan.

Line 255: be specific, how consistent, and stable? Over how long of the duration?

Line 264: cats with “CE”

Line 265: why only mention IBD? In humans or cats?

Line 269: missing reference.

Line 275: FAs concentrations in serum? Feces? Or RBC membranes? Not clear. How large is the study sample size?

Line 289: be specific. “notable” what does it mean? How many fold differences? No overlapping between groups or statistically different? Same analytical method from your study and this study?

Line 291: Check the author’s name.

Line 313: Check the references.

Line 336: Check the references.

6. PLOS authors have the option to publish the peer review history of their article (what does this mean?). If published, this will include your full peer review and any attached files.

Reviewer #1: No

---

## [Author Response · Author response to Decision Letter 0]

5 May 2024

Response to Review Comments to the Author

Reviewer #1: I would like to express my gratitude to the authors for sharing their fascinating study. The paper delves into the fatty acid profiles of the RBC membrane in both healthy cats and those with chronic enteropathies, with a thorough and well-articulated description of the study populations and methods. The introduction and discussion are highly informative, providing valuable insights into the topic. However, the results may benefit from some refinement, and the references require meticulous editing. Any typographical or grammatical errors should be corrected.

R: The authors are very grateful for the reviewer comments that, we believe, truly improved the manuscript. The issues have been addressed (as follows) and the manuscript has been modified accordingly. 

General comments:

• Please reconcile CE or FCE throughout the manuscript

R: When referred to cats, the acronym FCE has been used in the whole manuscript

• Please reconcile PUFA w-3 or w-3 PUFA

R: w-3 (or w-6) PUFA have been included instead of PUFA w-3 (or w-6)

Detailed comments:

Line 1: Title: mixed case or sentence case?

R: The title has been modified. Also the running title has been amended accordingly

Abstract

The word count of the abstract exceeds the limit. Please carefully read the author's guidelines.

R: The abstract has been modified and now is within words

Line 30, Gas chromatography “with which detection device (UV, MS, or others)” was used to quantitatively analyze a cluster of 11 “FAs”,

R: The information required has been included in the manuscript in the M&M section. The authors prefer maintain only “Gas chromatography” in the abstract to respect the limit of 300 words.

Line 31, “Including lipid homeostasis and enzyme activity indexes” This part is confusing. Those are not directly measured by the GC, but a calculation based on the results. Make it clear.

R: The part has been modified for clarity

Line 34, “PUFA” should not be abbreviated without the full name if only appears once.

R: The comment has been addressed

Line 37, “subgroups” of CE. No paragraphs in the abstract.

R: The sentence has been modified including FRE, IBD and LGITL

Line 38, In “humans” and dogs

R: The sentence has been modified

Introduction

Line 80: UC and CD first appear.

R: The comment has been addressed

Line 84: Check the abbreviation and provide their full names if first appear.

R: The full names have been provided

Materials and methods

Line 107: be specific, which serum biochemistry profile

R: All the biochemical analysis have been specified 

Line 108: feline “serum” pancreatic lipase “i…”

R: Corrected

Lines 128-132: need more polish.

R: This part has been modified for clarity

Line 142: FAME?

R: The full name “Fatty Acid Methyl Esters” has been included

Lines 145-149: please explain the information in the parentheses.

R: This refers to the standard nomenclature C:D where C is the number of carbon atoms in the fatty acid and D is the number of double bonds in the fatty acid. This nomenclature is widely recognized and the authors would like to suggest to omit the explanation in the manuscript. However, a clarification has been made in the caption of the new figure (see following comments)

Line 150: explains which are SFA, MUFA, PUFA, w-6 and w-3. A table summarizing the information would be appreciated.

R: A summarizing table has been included in the manuscript

Line 162-163: Did you test the normality of your data? Which tests? Did you correct the p-values for multiple comparisons?

R: Normality was checked using the D’Agostino Pearson test and this information has been included. Correction was done for analysis on more than two groups were performed with post-hoc tests (Student-Newman-Keuls test or Dunn test), as already specified in the manuscript.

Results

Line 171: Basic information is lacking. Signalment for both groups. Move the S1 and S2 tables to the main text.

R: Supplementary tables have been moved into the manuscript

Line 174: Be specific “positively” what does it mean? Decreased of FCEAI? No clinical signs?

R: This information is already provided in the Material and methods section: “the remission was considered complete if clinical signs were resolved or the FCEAI score was reduced by ≥75% after three weeks of dietary therapy”. The authors believe that is more appropriate to kept this information in M&M

Line 191: PCA and heatmap might be good ways to visualize this dataset with multiple variables. Please add them for readers to better appreciate the changes.

R: The authors are grateful for the suggestion. PCA and heatmap have been included.

Line 197: Are these P values corrected for multiple comparisons? Please add adjusted P-values or Q-values. What are the units of these variables? “IQR” is a better abbreviation for interquartile range.

R: The p-value refers to analysis including post-hoc test (please see Statistical analysis section). IQR is now used instead of IQ.

Lines 201-207: Please add and make it clear that the FAs are from “RBC membranes” in the context and captions of tables and figures. For example, concentrations of DPA in RBC membranes. FAs can be measured in other biological matrixes as well. Some representative figures would help readers to appreciate the differences between groups. The quality of the supplementary figures is poor. I am not sure if it’s due to the end of the submission portal or from the authors.

R: As required, the notion that FA are from RBC membranes has been included for clarity. Representative figures (previously provided as Supplementary Information) have been moved into manuscript. The quality of figures has been fixed.

Line 206: Table 3 can be in the supplementary data or have some representative figures of some results.

R: As suggested, the Table 3 has been moved in Supplementary Information (renamed as S1 table)

Line 217: Hypocobalaminemic cats (n=?), the changes were compared to which population?

R: The information has been included and the sentence modified for clarity

Line 220: No difference of what?

R: The sentence has been amended for clarity

Lines 222-225: Please add the 95% confidence interval and the numbers of observations of each statistic. For example, xyz acid (r = 0.25, 95% CI [0.18-0.33], n = 65, p = 0.002)

R: 95% confidence interval and the numbers of observations of each statistic have been included

Discussion

While this journal does not impose any word limit, it is recommended to consider editing/removing certain portions of both mice and human studies. This would ensure that the content is concise and impactful, leading to better engagement and understanding among readers in veterinary medicine.

The authors would like to thank the reviewer for the constructive comment. The discussion has been amended accordingly.

Line 238: Kindly explain how metabolic differences between the two species.

R: please see response to the following comment

Line 243: Do you know if certain fatty acids are essential to add to cat’s diet according to NRC, AFFCO, or other authorities? What are the most common fatty acids found in their diet? Any differences between extruded diet and canned food?

R to Q related to Line 238 and Line 243: An explanation of metabolic differences (mainly focused on lipid metabolism) has been included together with FA requirements of cats (lines 447-477)

Line 245: cats “with CE”. Please reconcile throughout the whole manuscript.

R: When referred to cats, the acronym FCE has been used in the whole manuscript

Line 250-251: missing reference.

R: The reference has been included

Line 253: be specific, how long is the lifespan.

R: The required data has been included

Line 255: be specific, how consistent, and stable? Over how long of the duration?

R: It is generally believed that these cells keep their FA distribution throughout their life. This clarification has been included in the manuscript. The duration is related to the lifespan, that is also included as per your request (see previous response)

Line 264: cats with “CE”

R: corrected

Line 265: why only mention IBD? In humans or cats?

R: The author recognize that the sentence can lead to misunderstanding and the sentence has been corrected. The term IBD has been substituted by a more generic “chronic inflammatory gastrointestinal disorder”

Line 269: missing reference.

R: The reference has been included

Line 275: FAs concentrations in serum? Feces? Or RBC membranes? Not clear. How large is the study sample size?

R: The sentence has been modified for clarity

Line 289: be specific. “notable” what does it mean? How many fold differences? No overlapping between groups or statistically different? Same analytical method from your study and this study?

R: As required by the reviewer, the discussions have been shortened. During the editing this part has been removed

Line 291: Check the author’s name.

R: The author’s name seems correct

Line 313: Check the references.

R: The references seem fine

Line 336: Check the references.

R: As required by the reviewer, the discussions have been shortened. During the editing this part has been removed

---

## [Decision Letter · Decision Letter 1]

11 Jul 2024

Evaluation of the fatty acid-based erythrocyte membrane lipidome in cats with food responsive enteropathy, inflammatory bowel disease and low-Grade Intestinal T-cell lymphoma

PONE-D-24-03902R1

Dear Dr. Crisi,

We’re pleased to inform you that your manuscript has been judged scientifically suitable for publication and will be formally accepted for publication once it meets all outstanding technical requirements.

Kind regards,

Juan J Loor

Academic Editor

PLOS ONE

Additional Editor Comments (optional):

Reviewers' comments:

Reviewer's Responses to Questions

**Comments to the Author**

1. If the authors have adequately addressed your comments raised in a previous round of review and you feel that this manuscript is now acceptable for publication, you may indicate that here to bypass the “Comments to the Author” section, enter your conflict of interest statement in the “Confidential to Editor” section, and submit your "Accept" recommendation.

Reviewer #1: All comments have been addressed

2. Is the manuscript technically sound, and do the data support the conclusions?

Reviewer #1: Yes

3. Has the statistical analysis been performed appropriately and rigorously? 

Reviewer #1: Yes

4. Have the authors made all data underlying the findings in their manuscript fully available?

Reviewer #1: Yes

5. Is the manuscript presented in an intelligible fashion and written in standard English?

Reviewer #1: Yes

6. Review Comments to the Author

Reviewer #1: I would like to thank the authors for their hard work on revising the papers. They addressed all the questions, and the current revision is suitable for publication.

7. PLOS authors have the option to publish the peer review history of their article (what does this mean?). If published, this will include your full peer review and any attached files.

Reviewer #1: **Yes: **Chi-Hsuan Sung

---

## [Editor Report · Acceptance letter]

19 Jul 2024

PONE-D-24-03902R1 

PLOS ONE

Dear Dr. Crisi, 

I'm pleased to inform you that your manuscript has been deemed suitable for publication in PLOS ONE. Congratulations! Your manuscript is now being handed over to our production team.

Kind regards, 

on behalf of

Dr. Juan J Loor 

Academic Editor

PLOS ONE